# Dietary Effect of *Clostridium autoethanogenum* Protein on Growth, Intestinal Histology and Flesh Lipid Metabolism of Largemouth Bass (*Micropterus salmoides*) Based on Metabolomics

**DOI:** 10.3390/metabo12111088

**Published:** 2022-11-09

**Authors:** Pinxian Yang, Xiaoqin Li, Wenxiang Yao, Menglu Li, Yuanyuan Wang, Xiangjun Leng

**Affiliations:** 1National Demonstration Center for Experimental Fisheries Science Education, Shanghai Ocean University, Shanghai 201306, China; 2Centre for Research on Environmental Ecology and Fish Nutrition (CREEFN) of the Ministry of Agriculture and Rural Affairs, Shanghai Ocean University, Shanghai 201306, China; 3Shanghai Collaborative Innovation for Aquatic Animal Genetics and Breeding, Shanghai Ocean University, Shanghai 201306, China

**Keywords:** *Clostridium autoethanogenum*, largemouth bass, growth, intestinal healthy, lipid metabolomics

## Abstract

*Clostridium autoethanogenum* protein (CAP) is a new single-cell protein explored in aquatic feeds in recent years. This study investigated the dietary effects of CAP replacing fishmeal (FM) on the growth, intestinal histology and flesh metabolism of largemouth bass (*Micropterus salmoides*). In a basal diet containing 700 g/kg of FM, CAP was used to substitute 0%, 15%, 30%, 45%, 70% and 100% of dietary FM to form six isonitrogenous diets (Con, CAP-15, CAP-30, CAP-45, CAP-70, CAP-100) to feed largemouth bass (80.0 g) for 12 weeks. Only the CAP-100 group showed significantly lower weight gain (WG) and a higher feed conversion ratio (FCR) than the control (*p* < 0.05). A broken-line analysis based on WG and FCR showed that the suitable replacement of FM with CAP was 67.1–68.0%. The flesh n-3/n-6 polyunsaturated fatty acid, intestinal protease activity, villi width and height in the CAP-100 group were significantly lower than those in the control group (*p* < 0.05). The Kyoto Encyclopedia of Genes and Genomes analysis showed that the metabolic pathway in flesh was mainly enriched in the “lipid metabolic pathway”, “amino acid metabolism”, “endocrine system” and “carbohydrate metabolism”. In conclusion, CAP could successfully replace 67.1–68.0% of dietary FM, while the complete substitution decreased the growth, damaged the intestinal morphology and down-regulated the lipid metabolites.

## 1. Introduction

In recent years, the largemouth bass (*Micropterus salmoides*) industry has rapidly expanded in many countries, including China. As a typically carnivorous fish, largemouth bass has a high requirement for protein, especially for fish meal. Generally, the commercial feeds of largemouth bass contain high levels of fish meal of up to 40–50%, leading to a high feed cost. Therefore, developing new protein sources to decrease fish meal inclusion has become an urgent task. In largemouth bass, the replacement of fish meal with alternative proteins has been reported on fermented plant protein [1,2], single-cell protein [3] and poultry by-product meal [4]. However, some disadvantages have been found in these alternative proteins, including the imbalanced amino acid composition and high anti-nutritional factors in plant protein sources, and the potential safety problems, such as lipid oxidation and unwarranted source in animal protein sources.

In recent years, bacterial protein has attracted more attention in aquatic feed [5,6]. *Clostridium autoethanogenum* protein (CAP) is a novel bacterial protein produced by the fermentation of *Clostridium autoethanogenum*. In the fermentation process, the bacterial utilizes carbon monoxide from steel-making waste gas and ammonia to form bacterial protein, then the fermentation liquid is centrifuged and dried to obtain CAP [7]. At present, the relatively complete genome sequence of this bacterium has been obtained, and no toxic genes were found [8]. Compared with traditional plant and animal protein sources, CAP contains less anti-nutritional factors and lower Salmonella and biogenic amines content [9]. Several studies have reported the successful inclusion of CAP in aquatic diets. In juvenile largemouth bass (initial body weight of 17.75 g), the substitution of 63% dietary fish meal with CAP showed no adverse impacts on hepatic and hindgut histology [10]. In large-size largemouth bass with a body weight of 224 g, CAP could replace 150 g/kg of fish meal in a diet containing 350 g/kg of fish meal, without adverse effects on the growth performance, feed utilization and intestinal histology [11]. In Jian carp *(Cyprinus carpio var.* Jian), the replacement of soybean meal with CAP promoted the growth performance and antioxidant capacity without an obvious effect on the liver and midgut [12]. In addition, the successful inclusion of CAP in aquatic diets has also been reported in grass carp (*Ctenopharyngodon idellus*) [9], black sea bream (*Acanthopagrus schlegelii*) [13] and tilapia (*Oreochromis niloticus*) [14]. The suitable replacement level of dietary fish meal with CAP was 42.9% for largemouth bass [11] and 45% for white shrimp (*Litopenaeus vannamei*) [15]. However, the excessive inclusion of bacterial protein decreased the growth and feed intake of black sea bream [13] and white shrimp [15]; thus, it seems that CAP could not completely substitute dietary fish meal.

Previous studies have proved the feasibility of replacing fish meal with CAP in commercial diets for largemouth bass [16]. However, in commercial diets, there are various protein ingredients such as chicken meal, soybean protein concentrate, corn gluten, soybean meal besides fish meal, and these ingredients may interfere with the replacement of fish meal with CAP. Therefore, to fully understand the replacement of fish meal with CAP, fish meal was used as the only protein source in the present study, then, different proportions of dietary fish meal were replaced by CAP to investigate the effects on the growth, intestinal histology and flesh lipid metabolism of largemouth bass. The findings will direct the application of CAP in the diets of carnivorous fishes and promote the sustainable development of aquafeeds with less cost and less dependence on marine products.

## 2. Materials and Methods

### 2.1. Experimental Design

The basal diet was designed to contain 700 g/kg of fish meal, and then 0%, 15%, 30%, 45%, 70% and 100% of dietary fish meal was replaced by CAP based on iso-protein to form six diets, named as Con (control group), CAP-15, CAP-30, CAP-45, CAP-70 and CAP-100, respectively. The protein ingredients were ground, screened (60-mesh) and mixed with oil and water (27%). Then, the mixture was extruded to form sinking pellets with a diameter of 2.0 mm using a single-screw extruder (SLP–45; Chinese Fishery Machinery and Instrument Research Institute, Shanghai, China) at an extruding temperature of 85–90 °C. All diets were air-dried naturally and preserved at 4 °C until use. The diets’ formulation and proximate composition are shown in Table 1.

The CAP was provided by Beijing Shoulang New Energy Technology Co., Ltd., Beijing, China. The product contains 842.0 g/kg of crude protein, 19.0 g/kg of crude lipid and 32.7 g/kg of crude ash. As a light-yellow powder with a special smell, CAP can be directly added into feed as a protein source. Dietary amino acids and fatty acids compositions of CAP are shown in Table 2 and Table 3.

### 2.2. Experimental Fish and Feeding Management

Largemouth bass were supplied by a local aquaculture farm in Qingpu, Shanghai, China. The fish were fed with commercial diets for 10 weeks until the body weight reached about 80 g. Then, a total of 216 fish with an initial body weight of 80.0 ± 0.5 g were randomly allocated into 18 cage s (1.5 × 1.0 × 1.2 m), with 3 replicates (cage) per treatment and 12 fish per cage. The cages were hung in indoor cement pools without direct sunshine. During the feeding period, the six diets were fed to the fish three times daily (08:00, 13:00, 17:00), and the daily feeding rate was 2–2.5% of body weight. The feed intake was adjusted according to the feeding behavior and water temperature to ensure no feed residue was left 10 min after feeding. About one third of the cultured water was renewed with filtrated pond water, and the waste in the pools was siphoned out once a week. Water quality was monitored every day, and the water temperature, dissolved oxygen, pH and ammonia nitrogen levels were 25–30 °C, >5 mg/L, 7.5–8.0 and <0.2 mg/L, respectively. The feeding trial was conducted at the Binhai Aquaculture Station, Shanghai Ocean University, and lasted for 12 weeks.

### 2.3. Sample Collection

Prior to slaughter, all fish were deprived of diets for 24 h (starvation), then counted and bulk weighed to calculate survival, weight gain (WG), feed intake (FI) and feed conversion ratio (FCR). Five fish per cage were selected randomly to individually measure body weight and body length for the calculation of condition factor (CF). Three fish per cage were euthanized by an overdose of anaesthetic (MS–222), and clinical signs of death were ensured prior to sampling. The blood was drawn from the caudal vein and centrifuged at 4 °C for 10 min (3000 r/min) to collect the serum, then frozen at −80 °C for further analysis. Then, the three fish were dissected, and visceral, liver and intestinal lipid were weighed to calculate the hepatosomatic index (HSI), viscerosomatic index (VSI) and intestinal fat ratio (IFR). The anterior intestine (1 cm) was sampled for the measurement of morphological structures, and the remaining intestine was stored at −80 °C for the activity measurement of digestive enzyme. About 1 g of dorsal flesh from the left side of two fish per cage was collected and frozen in liquid N_2_ for the metabolomic assay, and the remaining flesh was stored at −80 °C until use. In addition, another three fish per cage were stored at −20 °C for the proximate composition analysis of whole fish. 

### 2.4. Analytical Methods

#### 2.4.1. Growth Indicators

WG (%) = 100 × [(final weight (g) − initial weight (g))/initial body weight (g)]

FCR = feed intake (g)/weight gain (g)

CF (g/cm^3^) = 100 × final body weight (g)/final body length (cm)^3^

Survival (%) = 100 × final number of fish/initial number of fish

VSI (%) = 100 × visceral weight (g)/body weight (g)

HSI (%) = 100 × hepatopancreas weight (g)/body weight (g)

IFR (%) = 100 × (intestinal fat weight [g]/body weight [g]) 

FI (g/d/fish) = total feed intake (g)/days (d)/number of test fish

#### 2.4.2. Proximate Composition of Diets and Whole Fish

Crude lipid, crude protein, moisture and crude ash contents were analyzed following the method of AOAC (2000a). The moisture and crude ash contents were determined by drying samples to a constant weight at 105 °C in an oven or by combusting samples at 550 °C for 6 h in a muffle furnace. The crude protein content was estimated with the Kjeldahl system method (2300 Auto analyser; FOSS Tecator, AB, Hoganas, Sweden), and the crude lipid was measured gravimetrically after extraction by chloroform-methanol. 

#### 2.4.3. Biochemical Analysis

The anterior intestine samples were thawed at 4 °C, homogenized with nine volumes (*w*/*v*) of ice-cold saline (0.86% NaCl) and centrifuged for 15 min (3000 r/min) at 4 °C. The supernatant was collected and preserved at 4 °C for the measurement of digestive enzyme activity and soluble protein in 24 h. Digestive enzyme indexes such as protease and lipase (LPS) were measured by commercial kits (Shanghai Haling Biotechnology Co., Ltd., Shanghai, China), following the protocols provided by the supplier.

The contents of albumin (ALB, bromocresol green method); total protein (TP, biuret method); triglyceride (TG, GPO-PAP method); cholesterol (CHO, CHOD-PAP method); glucose (GLU, glucose oxidase method); malondialdehyde (MDA, TBA method); and the activities of alkaline phosphatase (AKP, AMP method) and superoxide dismutase (SOD, BioTekmethod) were determined using the kits provided by Shanghai Haling Biotechnology Co., Ltd.

#### 2.4.4. The Histology of Anterior Intestine

The anterior intestine (1 cm) was immersed in Bouin’s solution for 24 h, and then transferred into 100% ethyl alcohol. The tissue was dehydrated in a series of alcohol solutions and embedded in paraffin. Then, sections (5 µm) were cut and stained with hematoxylin-eosin and sealed with a neutral gum. The morphological structures of the tissues as villus height and width were observed using an imaging microscope (Nikon YS100, Tokyo, Japan). The image was analyzed with the Image J14.0 image analysis software. The selected images contained complete villi and muscle thickness, while the number of complete villi was not less than 1/3 of the intestinal sections, and the goblet cells were counted under 100 times the field of view. Number of goblet cells: number of goblet cells per 200 µm length of mucosa. Villus height, villus width and muscle thickness parameters were measured (10 fields per individual sample) according to the procedures described by Escaffre et al. [17].

#### 2.4.5. Fatty Acid Composition of Flesh

The fatty acid was measured by the method of boron trifluoride methyl esterification. The extracted fat was dissolved in 1 mL of hexane, then 2 mL of 14% boron trifluoride methanol solution was added. After a water bathing of 25 min at 100 °C (the first step of methyl esterification), benzene (2 mL) and methanol solution (2 mL) were added for another water bath (100 °C, 25 min) (the second step of methyl esterification). Then, the samples were mixed with distilled water and n-hexane. After centrifuging at 3000 r/min for 10 min, the supernatant was collected for fatty acid analysis by using an Agilent Technologies 7890B GS Syetem GC/MS (Agilent, Santa Clara, CA, USA). 

#### 2.4.6. Non-Targeted Metabolomic Analysis

The control, CAP-30 and CAP-70 groups were used for the metabolomic analysis. Flesh sample was precisely weighed (50 mg), then extracted with a mixture (400 µL) of methanol and water (4:1, *v*/*v*). The sample was treated by a tissue crusher (high-throughput tissue crusher Wonbio-96c, Wanbo Biotechnology Co., Ltd., Shanghai, China) at 50 Hz for 6 min, then subjected to cryogenic sonication treatment for 30 min, kept at −20 °C for 1 h and centrifuged (13,000× *g*) at 4 °C for 15min. The supernatant (20 µL) was collected and transferred for LC-MS/MS analysis by Shanghai Majorbio Bio-Pharm Technology Co., Ltd. (Shanghai, China).

The chromatographic column was ACQUITY UPLC HSS T3 (100 mm × 2.1 mm i.d., 1.8 µm; Waters, Milford, MA, USA). The mobile phases consisted of solvent A (0.1% formic acid) and solvent B (acetonitrile:isopropanol = 1:1 (*v*/*v*) containing 0.1% formic acid). The sample injection volume was 10 µL, and the flowing rate was 0.4 mL/min with a column temperature of 40 °C. Electrospray positive ion (ESI+) mode and electrospray negative ion (ESI−) mode were used to collect the mass spectrum signal (Triple TOFTM5600+, AB Sciex, San Diego, CA, USA). During the period of analysis, all samples were stored at 4 °C, and a quality control (QC) sample was inserted every 5–15 samples to evaluate the stability of the analytical system and assess the reliability of the results. 

Metabolism raw data was imported into Progenesis QI (Waters Corporation, Milford, USA) for preprocessing. Statistically significant metabolites among groups were selected with VIP > 1 and *p* < 0.05 for PCA. A partial least squares discriminate analysis (PLS-DA) was used for statistical analysis to determine flesh metabolic changes between comparable groups. The model validity was evaluated from model parameters R2 and Q2, which provided information for the interpretability and predictability to avoid the risk of over-fitting. Differential metabolites between two groups were summarized and mapped into their biochemical pathways through a metabolic enrichment and pathway analysis based on the KEGG (Kyoto Encyclopedia of Genes and Genomes) database search (http://www.kegg.com/ accessed on 3 February 2022) [18]. 

### 2.5. Statistical Analysis

The experimental data were presented as the mean and standard deviation (±SD). The statistical analysis was performed using the Statistical Package for the Social Sciences (SPSS) 19.0 for Windows (SPSS, Chicago, IL, USA). A one-way analysis of variance (ANOVA) was combined with the LSD method for multiple comparisons. A follow-up trend analysis was performed using orthogonal polynomial contrast to determine whether significant effects were linear and/or quadratic. The significance level for the differences among treatments was *p* < 0.05. In addition, a nonlinear regression analysis (binomial method) was used, and the CAP protein concentration data were used for binomial transformation and curve estimation.

## 3. Results

### 3.1. Growth Performance

In Table 4, the CAP-15, CAP-30, CAP-45 and CAP-70 groups showed the similar WG, FI and FCR to the control (*p* > 0.05), while the WG of CAP-100 group was decreased by 31.8%, and FCR was increased by 0.35 when compared to the control (*p* < 0.05). The VSI of CAP-45, CAP-70 and CAP-100 groups, and the HSI and CF of CAP-100 group were significantly lower than those of the control (*p* < 0.05). The broken-line model based on WG and FCR showed that the proper replacement ratio of dietary fish meal by CAP was 67.1% and 68.0%, equal to 374.6 g/kg and 380.3 g/kg of CAP inclusion (Figure 1a,b), respectively. The binomial regression analysis showed that the CAP protein concentration had a good direct fit with the WG and FCR of largemouth bass, and the optimal CAP concentrations for WG and FCR were 146 g/kg and 181 g/kg, respectively (Figure 2a,b).

### 3.2. Body Composition and Nutrients Retention

In the proximate composition of whole-body, there were no significant differences in crude ash, crude lipid and protein retention among all the groups (*p* > 0.05). The crude protein in the CAP-45, CAP-70 and CAP-100 groups, and the moisture in the CAP-70 and CAP-100 groups were significantly higher, while the lipid retention in the CAP-70 group was significantly lower than that in the control (*p* < 0.05) (Table 5).

### 3.3. Serum Biochemical Indices Analyses

In Table 6, no significant differences were detected in AKP activity, MDA, TG, TP, ALB and GLU contents in the CAP groups (*p* > 0.05) when compared to the control group (*p* > 0.05). CAP-30 and CAP-45 groups showed significantly higher SOD activity than the control (*p* < 0.05).

### 3.4. Intestinal Digestive Enzymes

Compared to the control group, the intestinal LPS activity was significantly increased in the CAP-45 group, while the protease activity was significantly decreased in the CAP-100 group (*p* < 0.05) (Table 7). 

### 3.5. The Histology of Anterior Intestine

As shown in Table 8, the villus height of the CAP-100 group and the villus width of the CAP-70 and CAP-100 groups were significantly lower than those of the control group (*p* < 0.05). The muscle thickness did not reveal any significant difference among the treatments (*p* > 0.05). 

When the replacement ratio of fishmeal with CAP reached 70% (CAP-70 and CAP-100 group), some goblet cells were observed to be enlarged, and some intestinal structure was damaged with shed and ruptured villus. The length and width of the intestinal villi tended to decrease with the increasing CAP inclusion (Figure 3). 

### 3.6. Fatty Acid Composition in Flesh

In Table 9, flesh SFAs in the CAP-30, CAP-45 and CAP-70 groups were significantly lower, while C20:4 were higher than those of the control group (*p* < 0.05). CAP-70 and CAP-100 groups showed significantly higher C20:1 than the control (*p* < 0.05). In addition, the n-6PUFAs of the CAP-70 group diet and the n-3/n-6 PUFAs of the CAP-100 group were significantly lower than those of the control (*p* < 0.05).

### 3.7. Flesh Metabolite Profiles

The PCA and PLS-DA models of flesh metabolites were performed to identify changes in metabolites in the control, CAP-30, CAP-70 and CAP-100 groups (Figure 4). 

The PLS–DA scores plot of the three groups showed strong clustering for flesh metabolism products without any overlap. In the permutation test, the result showed good repeatability and predictability of the model. A total of 507, 1709 and 1838 different metabolites, including 25, 103 and 134 named metabolites, were identified between the CAP-30 group and the control, between the CAP-70 group and the control, and between the CAP-100 group and the control, respectively (Figure 5).

The above identified metabolites were assigned to the KEGG database. In the CAP-30 group, seven KEGG pathways were classified, and the top priority was “lipid metabolism”, followed by “amino acid metabolism”, “cancer: Overview”, “polyketide metabolism”, “biosynthesis of other secondary metabolites” and “folding, classification and degradation” (Figure 6a). In the CAP-70 group, 16 KEGG pathways were significantly enriched in “lipid metabolism”, “amino acid metabolism”, “cancer: Overview”, “digestive system”, “endocrine system”, “carbohydrate metabolism”, “other amino acid metabolism” and “nucleotide metabolism” (Figure 6b). In the CAP-100 group, 27 KEGG pathway enrichment were observed as “lipid metabolism”, “cancer: Overview”, “amino acid metabolism”, “digestive system”, “endocrine system”, “carbohydrate metabolism”, “other amino acid metabolism”, “metabolism of vitamins”, etc. (Figure 6c). Among them, 2, 8 and 10 metabolites in the lipid metabolism pathway were detected in the groups of CAP-30, CAP-70 and CAP-100 when compared with the control group (Table 10).

## 4. Discussion

### 4.1. Effect of CAP on the Growth of Largemouth Bass

In this study, CAP successfully replaced dietary fish meal up to 70% without significant effects on the growth performance, and the low inclusion of CAP (146 g/kg) even numerically increased WG and decreased FCR. The fish meal replacement level obtained from the binomial regression analysis was lower than that obtained from the broken-line analysis (Figure 1 and Figure 2). Generally, the aim of replacing dietary fish meal is to produce a growth performance that is no lower than the high fish meal group, rather than better than the high fish meal group. Thus, the proper replacement level of dietary fish meal was suggested to be 67.1–68.0% in the present study. The complete substitution significantly decreased the weight gain and feed intake. Fish meal contains many active substances and unknown growth factors, such as small peptides, taurine, trimethylamine oxide (TMAO), etc., which are deficient in bacterial protein ingredients [19,20]. Small peptides could promote the intestinal tract movement, the transfer and absorption of amino acids, and increase the apparent digestibility [21]. Taurine can improve the utilization of feed and enhance the immune capacity of the fish [22,23]. TMAO can stimulate the fish to feed [24]. In addition, some essential amino acids in CAP are lower than those in fish meal, such as arginine (Table 2). The lack of arginine in diet has been reported to induce the erosion of fin and increase the mortality of channel catfish (*Ictalurus punctatus*) [25]. Meanwhile, the lysine content in CAP diets was relatively higher, which might aggravate the antagonism between lysine and arginine. In black sea bream [26], Indian major carp (*Labeo rohita*) [27] and Atlantic salmon [28], the imbalanced ratio of lysine to arginine in feed was found to decrease the growth. Furthermore, the existence of bacterial cell walls also reduces the nutrients’ digestibility. Therefore, the supplementation of exogenous amino acid, especially arginine, should be considered to balance the amino acid composition when CAP is included in diets. 

### 4.2. Effect of CAP on Serum Biochemical Indexes of Largemouth Bass

Serum TP, ALB, TG, CHO and glucose can reflect the metabolism, nutritional status and health of the fish. In the present study, the serum TP content was not affected by dietary CAP, but the serum ALB contents of the CAP-15, CAP-30 and CAP-45 group were significantly higher than that of the control group. In black sea bream [13] and Jian carp [12], replacing dietary fish meal with CAP had no significant effect on serum TP and ALB contents. Usually, the blood glucose content regulated by the nervous and endocrine systems is a dynamic equilibrium, which plays a pivotal role in maintaining the normal activities of fish. No significant difference was observed in serum GLU content among all the groups in this study. Similarly, the replacement of dietary fish meal with CAP (200 g/kg) showed no significant effect on the serum GLU content of Jian carp, but significantly reduced the contents of TG and CHO [12]. In black sea bream, the serum GLU, TG and CHO contents were also not affected by the replacement of fish meal (58.2%) with CAP [13]. Therefore, it seems that the dietary inclusion of CAP has a less negative effect on the nutritional metabolism of largemouth bass.

### 4.3. Effects of CAP on Digestive Ability and Intestinal Structure of Largemouth Bass

The height and length of intestinal villus can reflect the tissue structure of the intestine, which is positively correlated with the absorption capacity of intestine. In the present study, the complete replacement of fish meal with CAP damaged the intestinal structure with lowered intestinal villus width, height and ruptured villi. Similarly, an intestinal injury was also observed in grass carp, including microvilli shedding, lamina propria loosening and goblet cell reduction, when 10% CAP was used to substitute dietary soybean meal. However, the dietary inclusion of 200 g/kg of CAP showed no pathological damage to the intestine of Jian carp [12], and the replacement of 58.20% of fish meal with CAP did not negatively affect the digestive enzymatic activity of black sea bream [13]. The decreased digestive function and damaged intestinal structure of largemouth by the complete substitution of fish meal bass might be related to the lack of some active substances. The intestinal absorption capacity is related to the intestinal structure, and the decrease in intestinal digestive enzyme activity may be related to the decrease in the height and width of intestinal villi [18]. Some small peptides can stimulate the feed intake and maintain the structure of intestinal epithelium. Taurine can help the digestion and absorption of neutral fats, cholesterol, fat-soluble vitamins and other fat-soluble substances, as well as improve the digestibility of protein [29]. The supplementation of arginine in diets improved the digestion and absorption capacity of red drum (*Sciaenops ocellatus*) [30] and grouper (*Epinephelus coioides*) [31], and dietary arginine also increased the height of intestinal folds and improved the intestinal health of red drum [30].

### 4.4. Effects of CAP on Flesh Fatty Acid and Lipid Metabolism of Largemouth Bass

In addition to supplying energy, the fatty acids, phospholipids, steroids and their derivatives also play important roles in cell proliferation, apoptosis, immunity and inflammation [32]. Generally, the dietary proportion of fatty acids affects the fatty acid composition in flesh [33,34]. Although fish oil was added in fish meal substituted diets to reach the same lipid level as the control group, the content of n-3 PUFAs and n-3/n-6 PUFAs tended to decrease as the proportion of CAP increased (Table 3). In particular, the highest replacement (CAP-100 group) significantly decreased the n-3/n-6 PUFAs (Table 9). The low PUFAs content in CAP may lead to the decrease in PUFAs content in flesh with the increase in the proportion of CAP replacing fish meal.

Compared to the control group, the most significant metabolic pathway in the CAP-30, CAP-70 and CAP-100 groups was “lipid metabolism” (Table 10). Eight different metabolites related to fat metabolism were found in the comparison of the CAP-70 group and control group, which were involved in the glycerolipid metabolism, linoleic acid metabolism and glycerophospholipid metabolism. Among them, three different metabolites (Acetylcholine, LysoPC (20:5(5Z,8Z,11Z,14Z,17Z)), LysoPC (22:5(7Z,10Z,13Z,16Z,19Z)) were down-regulated. In the CAP-100 vs Con, 10 different metabolites were found to be involved in arachidonic acid, glycerophospholipid metabolism, linoleic acid, linolenic acid and the PPAR signaling pathway, and six different metabolites (Acetylcholine, LysoPC (20:5 (5Z,8Z,11Z,14Z,17Z)), LysoPC (22:5 (7Z,10Z,13Z,16Z,19Z)) were down-regulated. Moreover, six different metabolites (Acetylcholine, Galactosylsphingosine, PC(18:3(6Z,9Z,12Z)/20:5(5Z,8Z,11Z,14Z,17Z)), LysoPC(20:5(5Z,8Z,11Z,14Z,17Z)), LysoPC(24:0),11b,17a,21-Trihydroxypreg-nenolone) were down-regulated and involved in the metabolism of glycerophospholipids, arachidonic acid metabolism, linoleic acid metabolism, α-Linolenic acid metabolism and sphingolipid metabolism (Table 10). Linolenic acid belongs to the n-3 PUFAs family, and linoleic acid and arachidonic acid belong to the n-6 PUFAs family. The n-3 and n-6 PUFAs are necessary for the growth and development of fish, in which they can promote growth, lipid metabolism and immunity. In addition, the elevated n-3 PUFAs and n-3/n-6 PUFAs may play important anti-inflammatory and anti-cancer roles [35,36]. The decreased pathway of phospholipids may affect the function of receptors in the membrane and then impair the normal function of the cell [37]. This might be one reason that the immunity capacity was decreased in fish fed with the high FM substitution levels. Therefore, the decreased growth performance by the complete replacement of fish meal with CAP might be connected with the low polyunsaturated fatty acid content in muscle, the significant changes in lipid metabolism pathways and the down-regulation of some lipid metabolites. Further investigation is needed to focus on the pathways and genes involved in lipid metabolism. 

## 5. Conclusions

In the present study, CAP successfully replaced 70% dietary fish meal without significant effects on the growth and flesh quality of the largemouth bass. The complete replacement of fish meal with CAP reduced the weight gain, digestive ability, damaged the intestinal structure and down-regulated the lipid metabolites of largemouth bass.

## Figures and Tables

**Figure 1 metabolites-12-01088-f001:**
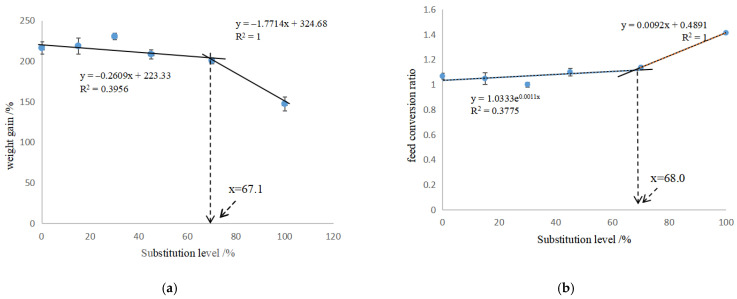
Broken-line analysis of weight gain (**a**) and feed conversion ratio (**b**) against substituted ratio of fish meal with CAP.

**Figure 2 metabolites-12-01088-f002:**
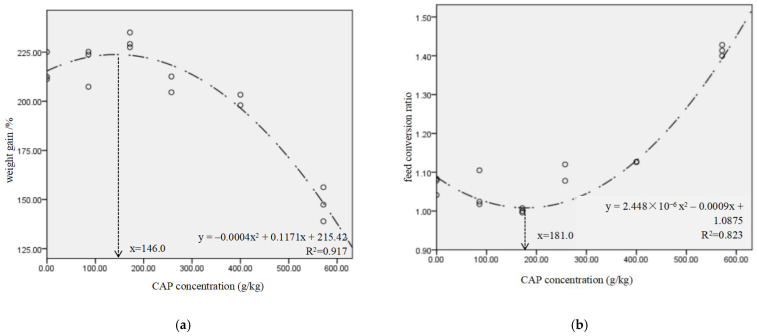
Relationship between CAP addition level and weight gain (**a**)/feed conversion ratio (**b**) for largemouth bass. Data are fitted with a nonlinear regression model (binomial regression model).

**Figure 3 metabolites-12-01088-f003:**
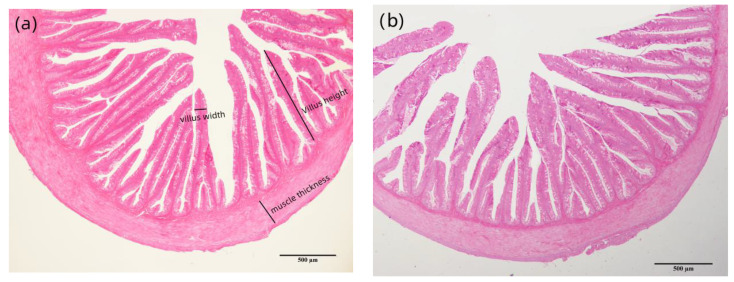
The foregut tissue structure of largemouth bass (40×). (**a**) Foregut tissue of control group; (**b**) foregut tissue of CAP-15 group; (**c**) foregut tissue of CAP-30 group; (**d**) foregut tissue of CAP-45 group; (**e**) foregut tissue of CAP-70 group; (**f**) foregut tissue of CAP-100 group. Note: The scale bar in the figure is the magnification contrast of micrograph.

**Figure 4 metabolites-12-01088-f004:**
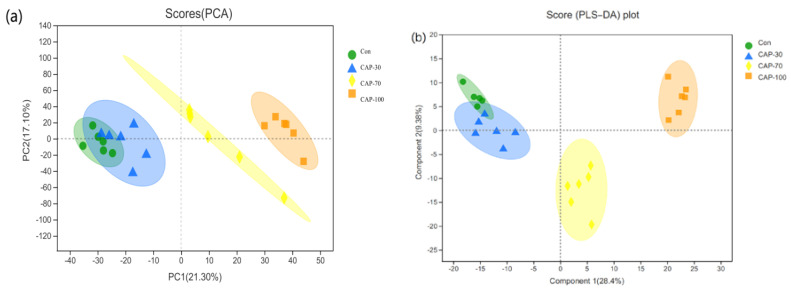
PCA score plots and PLS–DA score plots of muscle metabolomics. (**a**) PCA score plots of muscle metabolomics; (**b**) PLS–DA score plots of ESI+ mode and ESI− mode; (**c**) internal validation of PLS–DA score plots.

**Figure 5 metabolites-12-01088-f005:**
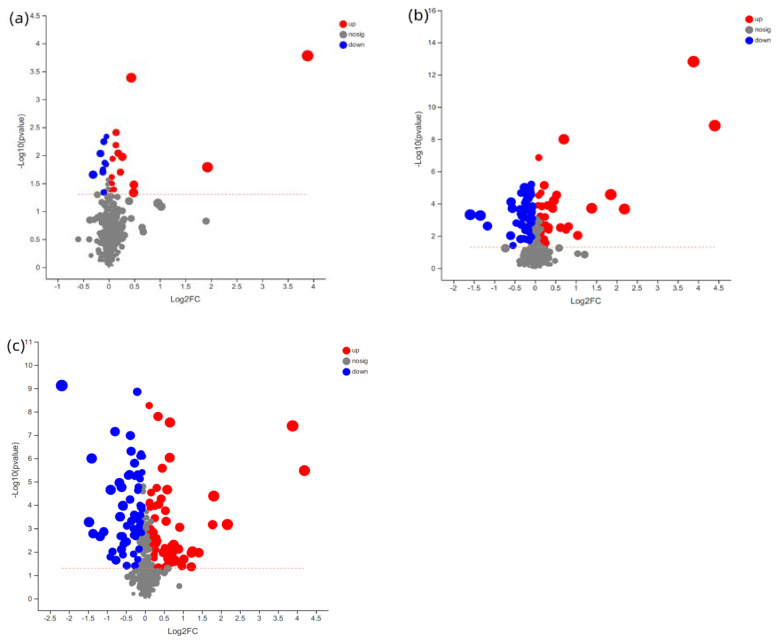
Volcano plots for the potential metabolomic features of muscle samples. (**a**) Volcano plots between CAP-30 and control group; (**b**) CAP-70 and control group; and (**c**) CAP-100 and control group.

**Figure 6 metabolites-12-01088-f006:**
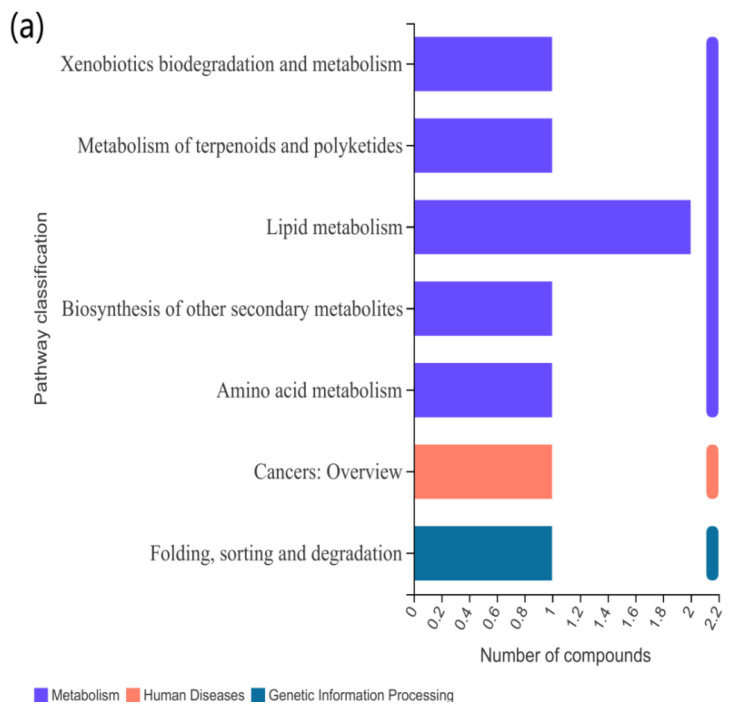
KEGG pathway classification of differently expressed metabolites. (**a**) KEGG pathway classification between CAP-30 and control group; (**b**) CAP-70 and control group; and (**c**) CAP-100 and control group.

**Table 1 metabolites-12-01088-t001:** Formulation and proximate composition of experimental diets (g/kg).

Ingredients ^1^	CON	CAP-15	CAP-30	CAP-45	CAP-70	CAP-100
Fish meal	700.0	595.0	490.0	385.0	210.0	0.0
CAP	0.0	85.80	171.6	257.3	400.0	571.6
Bone meal	0.0	15.0	30.0	45.0	69.0	98.0
Ca(H_2_PO_3_)_2_	19.0	19.0	19.0	19.0	19.0	19.0
Wheat flour	200.0	194.7	189.4	184.2	176.5	166.9
Fish oil	0.0	9.5	19.0	28.5	44.5	63.5
Soybean oil	25.0	25.0	25.0	25.0	25.0	25.0
Soybean lecithin	25.0	25.0	25.0	25.0	25.0	25.0
Choline chloride	1.0	1.0	1.0	1.0	1.0	1.0
Premix ^2^	10.0	10.0	10.0	10.0	10.0	10.0
Yeast extract	20.0	20.0	20.0	20.0	20.0	20.0
Total	1000.0	1000.0	1000.0	1000.0	1000.0	1000.0
Proximate composition						
Crude protein	503.8	498.8	498.9	500.0	498.2	497.8
Crude lipid	117.1	116.7	117.3	115.9	117.1	115.7
Crude ash	121.3	121.2	118.4	116.2	114.4	112.3
Moisture	81.4	81.6	80.9	81.1	79.7	78.1

^1^ The ingredients were purchased from the Yuehai Feed Company (Zhejiang, China), and the protein contents of the ingredients are as follows: fish meal (682.0 g/kg), wheat flour (117.0 g/kg), yeast extract (216.0 g/kg). ^2^ Premix provided the following per kg of diets: Lascorbate-2-mon-phosphate (35%), 600 mg; vitamin E, 300 mg; inositol, 200 mg; niacinamide, 80 mg; calcium pantothenate, 40 mg; vitamin A, 20 mg; vitamin D_3_, 10 mg; vitamin K_3_, 20 mg; vitamin B_1_, 10 mg; vitamin B_2_, 15 mg; vitamin B_6_, 15 mg; vitamin B_12_, 8 mg; biotin, 2 mg; wheat middling, 220 mg; zeolite, 332 mg; Fe, 300 mg; Zn, 200 mg; NaCl 100 mg; Cu, 30 mg; Mn, 25 mg; CoCl_2_·6 H_2_O (10%Co), 5 mg; Na_2_SeO_3_ (10%Se), 5 mg; KIO, 3 mg.

**Table 2 metabolites-12-01088-t002:** Amino acid composition of experimental diets (dry matter basis, g/kg).

Amino Acid	CON	CAP-15	CAP-30	CAP-45	CAP-70	CAP-100
Essential amino acid						
Lys	37.6	39.5	41.5	43.5	46.7	50.6
Met	14.6	14.4	14.2	14.0	13.7	13.4
Arg	30.3	28.9	27.4	26.0	23.7	20.8
His	15.2	14.5	13.7	13.0	11.7	10.2
Phe	26.2	25.2	24.3	23.3	21.7	19.8
Trp	5.4	5.2	4.96	4.8	4.4	4.0
Val	24.9	26.0	27.1	28.2	30.0	32.2
Ile	20.2	21.8	23.4	25.0	27.7	31.0
Leu	38.6	38.5	38.4	38.3	38.1	38.0
Thr	21.0	21.4	21.8	22.2	22.9	23.7
Non-essential amino acid						
Cys	21.0	20.1	19.1	18.2	16.6	14.7
Gly	4.6	4.7	4.9	5.0	5.2	5.5
Ser	20.0	21.6	23.3	24.9	27.6	30.8
Pro	43.8	45.6	47.3	49.1	51.9	55.4
Ala	5.4	5.2	5.0	4.8	4.4	4.0
Asp	24.9	26.0	27.1	28.2	30.0	32.2
Tyr	29.9	28.9	27.9	26.8	25.1	23.0
Glu	66.2	59.3	56.9	54.1	46.8	40
Total amino acids	449.8	446.8	448.26	449.4	448.2	449.3

Note: Glu, glutamic acid; Asp, aspartic acid; Leu, leucine; Ile, isoleucine; His, histidine; Gly, glycine; Thr, threonine; Ala, alanine; Arg, arginine; Phe, phenylalanine; Trp, tryptophan; Lys, lysine; Pro, proline; Tyr, tyrosine; Val, valine; Met, methionine; Ser, serine; Cys, cysteine.

**Table 3 metabolites-12-01088-t003:** Fatty acid composition of experimental diets (percentage of fatty acids, %).

Items	Con	CAP-15	CAP-30	CAP-45	CAP-70	CAP-100
C14:0	4.20	4.68	5.23	5.10	5.34	5.61
C15:0	0.46	0.55	0.66	0.75	0.92	1.12
C16:0	15.31	15.20	17.40	19.20	22.68	25.48
C17:0	0.71	0.61	0.52	0.42	0.27	0.08
C18:0	2.83	2.74	2.66	2.57	2.42	2.25
C20:0	0.29	0.25	0.21	0.17	0.11	0.03
SFAs	23.79	24.03	26.67	28.22	31.74	34.57
C16:1	3.90	3.85	3.79	3.74	3.64	3.54
C17:1	0.53	0.54	0.49	0.50	0.49	0.47
C18:1	14.18	15.23	15.28	14.96	15.92	15.86
C20:1	2.64	2.03	1.91	1.84	1.39	1.20
MUFAs	21.25	21.64	21.48	21.04	21.45	21.06
C16:2	0.24	0.25	0.21	0.25	0.19	0.21
C18:2	14.65	13.76	13.08	13.97	13.26	13.06
C20:2	3.01	2.92	2.91	2.66	4.07	3.68
C20:4	2.30	2.21	2.39	1.12	1.09	0.69
n-6 PUFAs	20.21	19.14	18.59	18.01	18.61	17.65
C18:3	2.83	2.58	2.31	2.21	1.99	1.38
C20:5	10.69	10.61	10.56	10.23	8.56	7.61
C22:5	10.83	10.41	10.00	9.78	9.46	9.34
C22:6	8.05	8.35	8.16	7.06	6.86	6.63
n-3 PUFAs	32.40	31.94	31.03	29.29	26.87	24.95
n-3/n-6	1.60	1.67	1.67	1.63	1.44	1.41

**Table 4 metabolites-12-01088-t004:** Effects of dietary *Clostridium autoethanogenum* protein on growth and body index of largemouth bass.

Parameters	Diets						Pr > F		
	Con	CAP-15	CAP-30	CAP-45	CAP-70	CAP-100	ANOVA	Linear	Quadratic
IBW/g	80.1 ± 0.2	80.0 ± 0.2	80.2 ± 0.2	80.1 ± 0.2	80.0 ± 0.2	80.2 ± 0.2	-	-	-
FBW/g	253.1 ± 6.2 ^ab^	255.1 ± 7.9 ^ab^	264.5 ± 3.3 ^a^	246.8 ± 4.5 ^b^	240.5 ± 3.1 ^b^	198.1 ± 6.9 ^c^	0.000	0.000	0.000
WG/%	216.3 ± 7.6 ^ab^	218.8 ± 9.9 ^ab^	230.5 ± 3.9 ^a^	208.6 ± 5.6 ^b^	200.6 ± 3.7 ^b^	147.5 ± 8.6 ^c^	0.000	0.000	0.000
FCR	1.06 ± 0.02 ^bc^	1.04 ± 0.04 ^bc^	1.00 ± 0.01 ^c^	1.10 ± 0.03 ^b^	1.13 ± 0.00 ^b^	1.41 ± 0.01 ^a^	0.000	0.000	0.000
FI/g/d/fish	2.20 ± 0.03 ^a^	2.18 ± 0.01 ^a^	2.18 ± 0.03 ^a^	2.15 ± 0.04 ^a^	2.19 ± 0.7 ^a^	1.99 ± 0.10 ^b^	0.002	0.001	0.006
CF/g/cm^3^	2.43 ± 0.07 ^abc^	2.55 ± 0.05 ^a^	2.49 ± 0.15 ^abc^	2.35 ± 0.03 ^bc^	2.28 ± 0.12 ^cd^	2.16 ± 0.02 ^d^	0.002	0.000	0.025
VSI/%	8.09 ± 0.17 ^a^	8.12 ± 0.18 ^a^	7.92 ± 0.10 ^ab^	7.65 ± 0.19 ^b^	7.70 ± 0.18 ^b^	7.78 ± 0.28 ^ab^	0.024	0.005	0.173
HSI/%	2.59 ± 0.18 ^a^	2.48 ± 0.09 ^ab^	2.49 ± 0.09 ^ab^	2.35 ± 0.13 ^ab^	2.28 ± 0.10 ^ab^	2.16 ± 0.17 ^b^	0.016	0.001	0.520
MSI/%	1.56 ± 0.13 ^a^	1.56 ± 0.07 ^ab^	1.52 ± 0.07 ^ab^	1.45 ± 0.10 ^ab^	1.48 ± 0.07 ^ab^	1.39 ± 0.07 ^b^	0.202	0.017	0.774
SR/%	100.0	100.0	100.0	100.0	100.0	100.0	-	-	-

Note: Within the same row, values with different superscripts are significantly different (*p* < 0.05). Pr > F: significant probability associated with the F-statistic. The value in the table is the average number of standard deviations (*n* = 3). Abbreviations: IBW, initial body weight; FBW, final body weight; WG, weight gain; FCR, feed conversion ratio; FI, feed intake; CF, condition factor; HSI, hepatosomatic index; VSI, viscerosomatic index.

**Table 5 metabolites-12-01088-t005:** Effects of dietary *Clostridium autoethanogenum* protein on proximate composition of whole body (fresh weight) and nutrients retention of largemouth bass.

Parameters	Diets						Pr > F		
	Con	CAP-15	CAP-30	CAP-45	CAP-70	CAP-100	ANOVA	Linear	Quadratic
Moisture (g/kg)	686.3 ± 4.1 ^b^	693.1 ± 3.8 ^ab^	694.3 ± 2.7 ^ab^	688.4 ± 5.1 ^b^	699.9 ± 4.5 ^a^	703.1 ± 2.1 ^a^	0.157	0.033	0.230
Crude protein (g/kg)	165.9 ± 6.2 ^b^	168.3 ± 3.1 ^ab^	172.9 ± 5.3 ^a^	171.5 ± 0.9 ^a^	172.7 ± 5.9 ^a^	170.1 ± 4.1 ^ab^	0.170	0.067	0.066
Crude lipid (g/kg)	56.1 ± 1.9	55.7 ± 0.5	50.8 ± 3.6	51.1 ± 2.9	51.2 ± 1.4	55.3 ± 2.7	0.032	0.173	0.027
Crude ash (g/kg)	43.4 ± 2.1	43.3 ± 3.0	47.0 ± 1.4	43.6 ± 1.5	46.4 ± 0.8	44.7 ± 2.5	0.096	0.132	0.322
Protein retention (%)	34.6 ± 0.1	34.6 ± 0.1	36.5 ± 0.1	37.2 ± 0.2	36.4 ± 0.1	35.6 ± 0.2	0.000	0.000	0.000
Lipid retention (%)	51.6 ± 2.2 ^a^	50.3 ± 1.4 ^ab^	45.3 ± 1.2 ^ab^	47.7 ± 0.6 ^ab^	43.8 ± 5.4 ^b^	51.7 ± 4.5 ^a^	0.000	0.067	0.001

Note: Within the same row, values with different superscripts are significantly different (*p* < 0.05).

**Table 6 metabolites-12-01088-t006:** Effects of dietary *Clostridium autoethanogenum* protein on serum biochemical indices of largemouth bass.

Parameters	Diets						Pr > F		
	Con	CAP-15	CAP-30	CAP-45	CAP-70	CAP-100	ANOVA	Linear	Quadratic
MDA (nmol/mL)	161.1 ± 16.9	162.9 ± 1.3	150.1 ± 13.9	158.5 ± 12.0	170.8 ± 12.2	168.6 ± 2.1	0.566	0.313	0.295
SOD (U/mL)	146.9 ± 13.6 ^b^	145.1 ± 5.1 ^b^	164.0 ± 1.7 ^a^	165.0 ± 7.1 ^a^	144.1 ± 7.1 ^b^	137.8 ± 3.7 ^b^	0.016	0.239	0.012
AKP (U/mL)	123.4 ± 2.8 ^ab^	129.7 ± 2.5 ^a^	128.1 ± 2.5 ^a^	125.0 ± 3.6 ^ab^	120.0 ± 7.2 ^ab^	110.7 ± 5.2 ^b^	0.000	0.000	0.000
TP (gprot/L)	52.5 ± 3.8	53.7 ± 6.1	52.8 ± 2.3	53.33 ± 1.5	52.2 ± 4.0	50.5 ± 1.0	0.952	0.500	0.483
ALB (gprot/L)	19.1 ± 2.2 ^ab^	20.2 ± 0.4 ^a^	19.9 ± 0.9 ^a^	20.9 ± 0.9 ^a^	17.1 ± 0.9 ^b^	17.2 ± 1.5 ^b^	0.021	0.025	0.033
GLU (mmol/L)	7.30 ± 0.43 ^ab^	8.16 ± 0.15 ^a^	8.12 ± 0.72 ^a^	7.86 ± 0.61 ^ab^	7.13 ± 0.46 ^b^	7.30 ± 0.28 ^ab^	0.129	0.236	0.067
TG (mmol/L)	17.7 ± 0.6	18.3 ± 0.5	17.0 ± 0.9	18.4 ± 0.5	18.3 ± 0.9	18.3 ± 1.1	0.090	0.169	0.020
CHO (mmol/L)	13.6 ± 1.2 ^ab^	14.2 ± 0.2 ^a^	13.1 ± 0.9 ^ab^	13.0 ± 1.4 ^ab^	12.8 ± 0.7 ^ab^	11.6 ± 0.8 ^b^	0.259	0.028	0.443

Note: Within the same row, values with different superscripts are significantly different (*p* < 0.05). Abbreviations: MDA, malondialdehyde; SOD, superoxide dismutase; AKP, alkaline phosphatase; TP, total protein; ALB, albumin; GLU, glucose; TG, triglyceride; CHO, cholesterol.

**Table 7 metabolites-12-01088-t007:** Effects of dietary *Clostridium autoethanogenum* protein on digestive enzymes activity of largemouth bass.

Parameters	Diets						Pr > F		
	Con	CAP-15	CAP-30	CAP-45	CAP-70	CAP-100	ANOVA	Linear	Quadratic
LPS (U/g prot)	356.4 ± 8.8 ^bc^	365.5 ± 15.4 ^b^	367.0 ± 12.1 ^ab^	386.3 ± 5.2 ^a^	364.7 ± 8.0 ^b^	342.3 ± 6.8 ^c^	0.010	0.769	0.000
Protease (U/mg prot)	2074.4 ± 32.5 ^a^	2019.7 ± 98.3 ^ab^	2006.4 ± 92.8 ^ab^	2003.2 ± 76.8 ^ab^	1965.1 ± 89.7 ^ab^	1909.6 ± 62.1 ^b^	0.265	0.023	0.062

Note: Within the same row, values with different superscripts are significantly different (*p* < 0.05).

**Table 8 metabolites-12-01088-t008:** Effects of dietary *Clostridium autoethanogenum* protein on intestinal structure of largemouth bass (μm).

Parameters	Diets						Pr > F		
	Con	CAP-15	CAP-30	CAP-45	CAP-70	CAP-100	ANOVA	Linear	Quadratic
Villus height	1201.5 ± 72.9 ^a^	1192.7 ± 36.8 ^a^	1256.9 ± 39.8 ^a^	1240.2 ± 66.1 ^a^	1129.3 ± 37.7 ^ab^	939.9 ± 52.3 ^b^	0.000	0.001	0.000
Villus width	105.24 ± 5.9 ^a^	110.1 ± 4.6 ^a^	106.5 ± 3.7 ^a^	103.4 ± 2.8 ^a^	96.9 ± 3.5 ^b^	97.1 ± 5.1 ^b^	0.016	0.002	0.132
Muscle thickness	222.2 ± 12.2	191.5 ± 7.6	198.5 ± 5.6	213.5 ± 11.8	211.2 ± 13.4	198.5 ± 8.1	0.050	0.216	0.798

Note: Within the same row, values with different superscripts are significantly different (*p* < 0.05).

**Table 9 metabolites-12-01088-t009:** Effects of dietary *Clostridium autoethanogenum* protein on muscle fatty acid composition of largemouth bass (percentage of fatty acids, %).

Parameters	Diets						Pr > F		
	Con	CAP-15	CAP-30	CAP-45	CAP-70	CAP-100	ANOVA	Linear	Quadratic
C14:0	2.03 ± 0.07	1.93 ± 0.06	1.89 ± 0.09	1.90 ± 0.06	1.91 ± 0.07	2.09 ± 0.14	0.268	0.621	0.034
C16:0	12.0 ± 0.9	11.6 ± 0.2	11.1 ± 0.2	10.7 ± 0.6	10.4 ± 0.3	11.9 ± 0.7	0.154	0.290	0.032
C18:0	4.26 ± 0.05	4.11 ± 0.18	4.02 ± 0.24	4.13 ± 0.16	4.08 ± 0.07	4.42 ± 0.10	0.703	0.645	0.251
SFA	18.3 ± 0.9 ^a^	17.7 ± 0.1 ^ab^	16.9 ± 0.4 ^b^	16.7 ± 0.6 ^b^	16.7 ± 0.5 ^b^	18.3 ± 0.8 ^a^	0.135	0.338	0.0.24
C16:1	5.49 ± 0.35	4.82 ± 0.40	5.57 ± 0.40	5.61 ± 0.11	5.18 ± 0.17	5.44 ± 0.34	0.373	0.707	0.968
C18:1	19.1 ± 0.6	19.6 ± 0.6	19.1 ± 0.1	19.5 ± 1.1	19.1 ± 0.9	18.5 ± 0.8	0.877	0.466	0.470
C20:1	0.84 ± 0.05 ^b^	0.85 ± 0.03 ^b^	1.05 ± 0.4 ^ab^	0.99 ± 0.05 ^ab^	1.12 ± 0.10 ^a^	1.16 ± 0.08 ^a^	0.064	0.006	0.806
MUFA	25.5 ± 1.1	25.3 ± 1.3	25.7 ± 0.5	26.1 ± 1.6	25.3 ± 1.2	25.1 ± 1.1	0.950	0.919	0.427
C18:2	16.1 ± 0.4	16.3 ± 0.2	15.5 ± 0.6	15.9 ± 0.3	14.8 ± 0.5	15.9 ± 0.5	0.027	0.044	0246
C20:2	0.63 ± 0.05	0.63 ± 0.3	0.63 ± 0.4	0.60 ± 0.3	0.60 ± 0.10	0.61 ± 0.15	0.688	0.212	0.789
C20:3	1.65 ± 0.2	1.62 ± 0.2	1.63 ± 0.4	1.61 ± 0.02	1.62 ± 0.14	1.61 ± 0.02	0.617	0.164	0.601
C20:4	0.23 ± 0.01 ^b^	0.26 ± 0.01 ^ab^	0.35 ± 0.05 ^a^	0.34 ± 0.03 ^a^	0.31 ± 0.06 ^a^	0.29 ± 0.02 ^ab^	0.133	0.116	0.035
n-6PUFAs	18.6 ± 0.2 ^a^	18.8 ± 0.1 ^a^	18.1 ± 0.5 ^ab^	18.5 ± 0.2 ^ab^	17.3 ± 0.4 ^b^	18.3 ± 0.1 ^ab^	0.017	0.029	0.303
C20:5	3.54 ± 0.15	3.45 ± 0.25	3.22 ± 0.24	3.29 ± 0.21	2.83 ± 0.12	2.70 ± 0.04	0.208	0.023	0.622
C20:5	2.48 ± 0.15	2.36 ± 0.19	2.46 ± 0.20	2.42 ± 0.08	2.48 ± 0.21	2.31 ± 0.08	0.911	0.656	0.745
C22:6	20.1 ± 2.7	20.7 ± 1.5	20.4 ± 1.8	20.3 ± 1.7	19.4 ± 0.4	19.9 ± 0.8	0.975	0.612	0.776
n-3PUFAs	26.2 ± 2.3	26.5 ± 0.9	26.1 ± 1.8	26.0 ± 1.2	24.8 ± 0.3	24.9 ± 0.7	0.728	0.202	00617
n-3/n-6	1.43 ± 0.08 ^a^	1.40 ± 0.05 ^ab^	1.44 ± 0.06 ^a^	1.40 ± 0.05 ^ab^	1.43 ± 0.03^a^	1.33 ± 0.04 ^b^	0.874	0.554	0.386

Note: Within the same row, values with different superscripts are significantly different (*p* < 0.05). Abbreviations: SFA, saturated fatty acid; MUFA, monounsaturated fatty acid; PUFAs, polyunsaturated fatty acid.

**Table 10 metabolites-12-01088-t010:** List of discriminating KEGG metabolites in CAP-30, CAP-70, CAP-100 and control groups.

Metabolite	Formula	FC	*p*. Value	KEGG Pathway Description
CAP-30 vs Con				
LysoPC(22:1(13Z))	C_30_H_60_NO_7_P	2.59	0.02	Glycerophospholipids
Cortolone	C_21_H_34_O_5_	1.99	0.02	Lipid metabolism; steroid hormone biosynthesis
CAP-70 vs Con				
Uridine diphosphate glucose	C_15_H_24_N_2_O_17_P_2_	1.47	0.01	Glycerolipid metabolism; amino sugar and nucleotide sugar metabolism
9,10-DHOME	C_18_H_34_O_4_	1.19	0.01	Linoleic acid metabolism
Cortolone	C_21_H_34_O_5_	1.16	0.01	Lipid metabolism; steroid hormone biosynthesis
LysoPC(22:1(13Z))	C_30_H_60_NO_7_P	1.29	0.01	Glycerophospholipid metabolism; choline metabolism in cancer
Eicosapentaenoic Acid	C_20_H_30_O_2_	1.03	0.03	Biosynthesis of unsaturated fatty acids
LysoPC(24:0)	C_32_H_66_NO_7_P	0.80	0.03	Glycerophospholipid metabolism; choline metabolism in cancer
LysoPC(20:5(5Z,8Z,11Z,14Z,17Z))	C_28_H_48_NO_7_P	0.95	0.00	Glycerophospholipid metabolism; choline metabolism in cancer
Acetylcholine	C_7_H_15_NO_2_	0.89	0.04	Glycerophospholipid metabolism; regulation of actin cytoskeleton
CAP-100 vs Con				
9,10-DHOME	C_18_H_34_O_4_	1.19	0.00	Lipid metabolism; linoleic acid metabolism
LysoPC(20:1(11Z))	C_28_H_56_NO_7_P	1.06	0.01	Glycerophospholipid metabolism; choline metabolism in cancer
9(S)-HODE	C_18_H_32_O_3_	1.10	0.00	Linoleic acid metabolism; PPAR signaling pathway
PC(22:6(4Z,7Z,10Z,13Z,16Z,19Z)/18:3(6Z,9Z,12Z))	C_48_H_78_NO_8_P	1.32	0.01	Glycerophospholipid metabolism; arachidonic acid metabolism; linoleic acid metabolism; α-Linolenic acid metabolism
Galactosylsphingosine	C_24_H_47_NO_7_	0.83	0.01	Sphingolipid metabolism
LysoPC(20:5(5Z,8Z,11Z,14Z,17Z))	C_28_H_48_NO_7_P	0.91	0.00	Lipid metabolism
11b,17a,21-Trihydroxypreg-nenolone	C_21_H_32_O_5_	0.94	0.00	Lipid metabolism; steroid hormone biosynthesis
LysoPC(22:5(7Z,10Z,13Z,16Z,19Z))	C_30_H_52_NO_7_P	0.93	0.01	Glycerophospholipid metabolism; choline metabolism in cancer
Acetylcholine	C_7_H_15_NO_2_	0.65	0.00	Glycerophospholipid metabolism; regulation of actin cytoskeleton
PC(18:3(6Z,9Z,12Z)/20:5(5Z,8Z,11Z,14Z,17Z))	C_46_H_76_NO_8_P	0.91	0.00	Glycerophospholipid metabolism; arachidonic acid metabolism; linoleic acid metabolism; α-Linolenic acid metabolism

## Data Availability

All data are contained within the article.

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
