# Peer review of "Dietary Effect of Clostridium autoethanogenum Protein on Growth, Intestinal Histology and Flesh Lipid Metabolism of Largemouth Bass (Micropterus salmoides) Based on Metabolomics"

_metabolites, 2022, doi:10.3390/metabo12111088_

Round 1

Reviewer 1 Report

General aspect

The abstract contemplates all experimental aspects, emphasizing the main results and is in accordance with the journal's norms. The introduction is well balanced in terms of the experimental problem and the importance of using the replacement of protein content in the diet of the species.

Methodology

The methodology poses serious questions about the data analysis strategy - at least there is some confusion in this regard.

2.4.3 there was no hematological analysis, but plasma biochemistry - replace and adjust the correct term.

2.4.4. Histology - correct 5mm by 5 µm cut.

Specify how the images were measured, clarifying number of histological sections/specimen and number of villi per section. Include how the goblet cells were measured (shown as enlarged - in the results).

Statistical procedure

If the authors intended to determine the estimated inflection point/dose/concentration, the curve generated by the polynomial model does not seem to be the most adequate. Therefore, I suggest that the data be analyzed by nonlinear regression analysis (binomial method), with the nominal protein concentration data (mg/Kg) logn-transformed, performing a logistic curve fit with a respective inflection point/dose/concentration.

Results

Fig 1. must contain the meaning of the abbreviations (WG and FCR). In addition, the fit of second-order polynomial curves (quadratics) must be presented. The presentation of these results is confusing. The SPSS provides good curve fitting and not two linear curves, as shown. Review and remodel this analysis and presentation of the results.

Images should present more clearly and with better resolution. Images are predominantly eosinophilic (red color) without demonstrating basophilic nuclei. It is suggested that higher magnification images, with more clarity in the structures, should be presented with better color definitions (H&E).

A complete description should be provided regarding comparing the set of the morphology of the histological sections. If there were damages to the tissue structure, it should be described or promptly elucidated which morphological elements were affected by the increase in concentration. Note that there is mention of "damage," "rupture villi," and "injury" in the discussion of these findings. These morphological aspects must be highlighted in the description of the results, even if subjectively. If not important, it should be taken from the context of the study.

Discussion

4.1. - 312-320 p. It is part of the literature review to support the study. It can be taken out of this context.

4.3. P. 357-358. The statement that the height and width of the villi with the intestinal absorptive capacity should be referenced.

359 p. There was a reduction in the height and width of the villi, but the damage caused by increased protein concentration was not reported in the results, such as villus disruption and goblet cell enlargement. The size of these findings should be promptly reported in the results.

Conclusion

Again, the intestinal damage structure was not reported in the results, but it seems essential for being at the conclusion. In addition, the morphology of intestinal samples should be better reported in the results.

Author Response

-Reviewer #1:

2.4.3 there was no hematological analysis, but plasma biochemistry - replace and adjust the correct term.

Response: theplasma biochemistry”has been revised as“Biochemical analysis” (Line 167)

2.4.4. Histology - correct 5mm by 5 µm cut.

Specify how the images were measured, clarifying number of histological sections/specimen and number of villi per section. Include how the goblet cells were measured (shown as enlarged - in the results).

Response:Thank you for your advice, the description of intestinal histological image measurements has been added in Line 188-190.

Statistical procedure

If the authors intended to determine the estimated inflection point/dose/concentration, the curve generated by the polynomial model does not seem to be the most adequate.

Therefore, I suggest that the data be analyzed by nonlinear regression analysis(binomial method), with the nominal protein concentration data (mg/Kg) logn-transformed, performing a logistic curve fit with a respective inflection point/dose/concentration.

Response:Thank you for your suggestion. Nonlinear regression analysis (binomial algorithm) has been used to compare CAP protein concentration with logarithmic conversion and weight gain/bait coefficient to obtain fitting data. Both analysis were kept, and the fish meal replacement level obtained from binomial regression analysis was lower than that obtained from broken line analysis, which has been discussed in Line 248-251.

Results

Fig 1. must contain the meaning of the abbreviations (WG and FCR). In addition, the fit of second-order polynomial curves (quadratics) must be presented. The presentation of these results is confusing. The SPSS provides good curve fitting and not two linear curves, as shown. Review and remodel this analysis and presentation of the results.

Response: Binomial regression curves were fitted by SPSS and added to Figure 2 a, d.

A complete description should be provided regarding comparing the set of the morphology of the histological sections. If there were damages to the tissue structure, it should be described or promptly elucidated which morphological elements were affected by the increase in concentration. Note that there is mention of "damage,""rupture villi," and "injury" in the discussion of these findings. These morphological aspects must be highlighted in the description of the results, even if subjectively. If not important, it should be taken from the context of the study.

Response: Histological sections were part of our focus, and the intestinal injury mentioned in the discussion is supplemented in Result 3.5 (Line 297-300).

Discussion

4.1. - 312-320 p. It is part of the literature review to support the study. It can be taken out of this context.

Response:Thank you for your suggestions. Relevant content has been added to the review (Line 61-70).

4.3. P. 357-358. The statement that the height and width of the villi with the intestinal absorptive capacity should be referenced.

Response:They have been added in line 401-403.

359 p. There was a reduction in the height and width of the villi, but the damage caused by increased protein concentration was not reported in the results, such as villus disruption and goblet cell enlargement. The size of these findings should be promptly reported in the results.

Response: Related result have been added to line 297-300.

Reviewer 2 Report

The current work addresses one of the major concerns in aquaculture nutrition nowadays, the replacement of fish meal for other equal nutritious sources of protein in fish feed. The work was focused on protein of bacterial origin, and an extensive evaluation of the changes at the molecular level were measured. Therefore, this work provides an extensive and interesting evaluation of biomarkers for antioxidant capacity, immune response and metabolism. The manuscript is easy to read, and some minor typing errors must be addressed (as in lines 173 or 184).

Author Response

-Reviewer #2:

The current work addresses one of the major concerns in aquaculture nutrition nowadays, the replacement of fish meal for other equal nutritious sources of protein in fish feed. The work was focused on protein of bacterial origin, and an extensive evaluation of the changes at the molecular level were measured. Therefore, this work provides an extensive and interesting evaluation of biomarkers for antioxidant capacity, immune response and metabolism. The manuscript is easy to read, and some minor typing errors must be addressed (as in lines 173 or 184).

Response: Thanks for the reviewer’s appreciation, and the wrong font has been corrected.

Reviewer 3 Report

Comments:

Abstract: Missing introductory statement. Authors need to say something about their study in brief form.

Ø  This study investigated dietary effects of Clostridium autoethanogenum protein (CAP) re-placing fishmeal (FM) on the growth, intestinal histology and flesh metabolism of largemouth bass (Micropterus salmoides)> protein will not be italic

Ø  SOD (Spell out), PUFAs (Spell out for the 1st time).

Ø  The serum SOD activity in CAP-30 and CAP-45 groups was significantly higher, while the flesh n-6 PUFAs content in CAP-70 group, flesh n-3/n-6 PUFAs, intestinal protease activity, villi width and height in CAP-100 group were significantly lower than those in the control group (P<0.05).> Confusing statement> Please break down it for better understanding.  

Ø  KEGG> Spell out.

Introduction: Please write about the outcome of this research, how it will help to fish industry and science or community etc.

Methods: Con> What is this? Spell out. Please explain about the CAP, how you get it, its shape and form and how you process it etc.

Ø  The six diets contained 700 g/kg, 595 g/kg, 490 g/kg, 385 g/kg, 210 g/kg, 0 g/kg fish

meal with the inclusion of 85.8 g/kg, 171.6 g/kg, 257.3 g/kg, 400 g/kg and 571.6 g/kg CAP,

respectively> It is not in correct for if you use “respectively, 1st one is six and 2nd one is seven. Please correct it.

Ø  Lines 94 and 95> If your CAP crude protein content of 842.0 g/kg, then I am not sure why in lower mixer like CAP 15% you have detected more protein than CAP 100% (Table 1). Any justification?

Ø  Table 2, Please give legend/spell out for Lys and others at below of the Table and all over the manuscript like TP, TG and so many for the 1st time.

Ø  2.4.1: What is CF? Spell out.

Ø  Line 210> P will be italic

Results

Line 264> The intestinal section is shown in Figure 2. Remove this line and refer Figure to in any statement or writing. Lines 281 and 281> Edit it as mentioned before.

Line 287> (Mahadevan, Shah, Marrie, Slupsky, 2008), Please check journal format.

Line 294> The above identified metabolites were assigned to the KEGG database> If I am not wrong, did not find any explanation about this KEGG in methods why authors need to do this?

 Discussion

4.1 Clostridium autoethanogenum> It will not be italic if journal style for this heading is italic. Please check it for all.

Line 373> epinephelus> Epinephelus

 So far others are acceptable in discussion part. However, authors need to polish their discussion part rather than comparing their fundings with others only.

Author Response

-Reviewer #3:

Abstract: Missing introductory statement. Authors need to say something about their study in brief form.

Response: Relevant information has been added (line 16-17).

Ø  This study investigated dietary effects of Clostridium autoethanogenum protein (CAP) re-placing fishmeal (FM) on the growth, intestinal histology and flesh metabolism of largemouth bass (Micropterus salmoides)> protein will not be italic

Response: The wrong font has been corrected (line 348, 374, 389, 411).

Ø  SOD (Spell out), PUFAs (Spell out for the 1st time).

Response: They have been added in line 24.

Ø  The serum SOD activity in CAP-30 and CAP-45 groups was significantly higher, while the flesh n-6 PUFAs content in CAP-70 group, flesh n-3/n-6 PUFAs, intestinal protease activity, villi width and height in CAP-100 group were significantly lower than those in the control group (P<0.05).> Confusing statement> Please break down it for better understanding.  

Response: The statement has been simplified for better understanding and for the word limit in ABSTRACT section.

Ø  KEGG> Spell out.

Response: They have been added in line 26.

Introduction: Please write about the outcome of this research, how it will help to fish industry and science or community etc.

 Response: They have been added in line 79-81.

Methods: Con> What is this? Spell out. Please explain about the CAP, how you get it, its shape and form and how you process it etc.

Response: They have been added in line 87.

Ø  The six diets contained 700 g/kg, 595 g/kg, 490 g/kg, 385 g/kg, 210 g/kg, 0 g/kg fish

meal with the inclusion of 85.8 g/kg, 171.6 g/kg, 257.3 g/kg, 400 g/kg and 571.6 g/kg CAP,

respectively> It is not in correct for if you use “respectively, 1st one is six and 2nd one is seven. Please correct it.

Response: They have been revised.

Ø  Lines 94 and 95> If your CAP crude protein content of 842.0 g/kg, then I am not sure why in lower mixer like CAP 15% you have detected more protein than CAP 100% (Table 1). Any justification?

 Response: We designed the six diets based on iso-protein. Actually, there may be some small differences in the measured values.

Ø  Table 2, Please give legend/spell out for Lys and others at below of the Table and all over the manuscript like TP, TG and so many for the 1st time.

 Response: They have been added in Table 2.

Ø  2.4.1: What is CF? Spell out.

 Response: They have been added in line 133.

Ø  Line 210> P will be italic

 Response: The wrong font has been corrected (Line 220).

Results

Line 264> The intestinal section is shown in Figure 2. Remove this line and refer Figure to in any statement or writing. Lines 281 and 281> Edit it as mentioned before.

Response: Revised in the paper.

Line 287> (Mahadevan, Shah, Marrie, Slupsky, 2008), Please check journal format.

Response: The wrong font has been deleted.

Line 294> The above identified metabolites were assigned to the KEGG database> If

I am not wrong, did not find any explanation about this KEGG in methods why authors need to do this?

Response:  KEGG is a comprehensive database widely used in omics research, which is widely used in the integration and interpretation of large-scale datasets obtained from genome sequencing and other high-throughput experimental techniques.Similar, The KEGG database is also used in the "Mechanism Analysis of Metabolic Fatty Liver on Largemouth Bass (Micropterus salmoides) Based on Integrated Lipidomics and Proteomics" article in this journal.

 Discussion

4.1 Clostridium autoethanogenum> It will not be italic if journal style for this heading is italic. Please check it for all.

Response: They have been revised.

Line 373> epinephelus> Epinephelus

Response: The wrong font has been corrected (line 408).

 So far others are acceptable in discussion part. However, authors need to polish their discussion part rather than comparing their fundings with others only.

Response: Thanks for your suggestions.We have added and revised the discussion.

Round 2

Reviewer 1 Report

Dear authors,

Some aspects of the text have been well-revised and are now ok. The specific places where the measurements were estimated should be shown in the intestine images. Include in the methodology how many replicates were made for the morphometric intestinal variable values. Explain in the methodology how the mucous cells were counted. These textual aspects must be considered to improve the manuscript. 

Author Response

Dear reviewers, thank you very much for your suggestions on the article, and look forward to your reply. We submit here the revised manuscript as well as a list of changes.

Some aspects of the text have been well-revised and are now ok. The specific places where the measurements were estimated should be shown in the intestine images. Include in the methodology how many replicates were made for the morphometric intestinal variable values. Explain in the methodology how the mucous cells were counted. These textual aspects must be considered to improve the manuscript.

Response: Thank you for your suggestions. The intestinal morphometric method has been added to the material method, and the specific places of the estimated measurements has been added to the picture (Line 186-189, and Figure 3 a).
